# Investigation of Synthesis, Characterization, and Finishing Applications of Spherical Al_2_O_3_ Magnetic Abrasives via Plasma Molten Metal Powder and Powder Jetting

**DOI:** 10.3390/mi15060709

**Published:** 2024-05-28

**Authors:** Shujun Wang, Yusheng Zhang, Shuo Meng, Yugang Zhao, Jianbing Meng

**Affiliations:** 1College of Agricultural Engineering and Food Science, Shandong University of Technology, Zibo 255049, China; 2School of Mechanical Engineering, Shandong University of Technology, Zibo 255049, China

**Keywords:** composites, Al_2_O_3_ magnetic abrasive, plasma molten metal powder, magnetic abrasive finishing, AZ31B alloy

## Abstract

Magnetic abrasive finishing (MAF) is an efficient finishing process method using magnetic abrasive particles (MAPs) as finishing tools. In this study, two iron-based alumina magnetic abrasives with different particle size ranges were synthesized by the plasma molten metal powder and powder jetting method. Characterization of the magnetic abrasives in terms of microscopic morphology, phase composition, magnetic permeability, particle size distribution, and abrasive ability shows that the magnetic abrasives are spherical in shape, that the hard abrasives are combined in the surface layer of the iron matrix and remain sharp, and that the hard abrasives combined in the surface layer of the magnetic abrasives with smaller particle sizes are sparser than those of the magnetic abrasives with larger particle sizes. The magnetic abrasives are composed of α-Fe and Al_2_O_3_; the magnetic permeability of the magnetic abrasives having smaller particle sizes is slightly higher than that of the magnetic abrasives having larger particle sizes; the two magnetic abrasives are distributed in a range of different particle sizes; the magnetic abrasives have different magnetic permeabilities, which are higher than those of the larger ones; both magnetic abrasives are distributed in the range of smaller particle sizes; and AZ31B alloy can obtain smaller surface roughness of the workpiece after the grinding process of the magnetic abrasives with a small particle size.

## 1. Introduction

With the rapid development of aerospace, biomedical, military, transportation, information communication, and other industries, the performance and service life of many basic parts with high hardness, complex surfaces, and other characteristics, such as being difficult-to-machine, put forward higher requirements [1,2,3,4,5,6,7]. The surface roughness of parts has a direct relationship with their performance and life; in general, the lower the surface roughness of the parts, the higher the performance and life of the parts [8]. Many scholars have discovered the potential of magnetic abrasive finishing technology in reducing the surface roughness of parts through research [9,10,11,12,13,14,15]. Magnetic abrasive finishing technology is a kind of finishing method that attracts a composite material composed of ferromagnetic metal powder and hard abrasive powder to be displaced relative to the surface of the part by the attraction of magnetic field, and it generates a small force [16]. Magnetic abrasive finishing technology is characterized by flexibility, adaptability, no need for trimming, low-temperature rise, and easy automation [16].

Magnetic abrasives are used as tools for magnetic abrasive finishing, and their performance directly affects the efficiency and processing quality of magnetic abrasive finishing [17]. At present, the methods of preparing magnetic abrasives include the mechanical mixing method [18], the bonding method [19], the sintering method [20], the plasma spraying method [21], the in situ alloy hardening method [22], the chemical plating method [23], the iron-based nitriding method [24], and the mixing powder atomization fast condensation method [25]. However, all of these magnetic abrasive preparation methods have some insurmountable disadvantages. The mechanical mixing method [18] is a method of magnetic abrasive obtained by adding oleic acid, transformer oil, and epoxy resin mixed in iron powder and hard abrasive powder. The magnetic abrasive prepared by this method in the magnetic abrasive finishing process will be separated due to the attraction of the magnetic poles and the iron powder and hard abrasive, and therefore the processing efficiency of magnetic abrasive finishing is greatly reduced. The bonding method [19] uses an adhesive to mix iron powder and hard abrasive powder proportionally to form magnetic abrasives. The process of producing magnetic abrasives by this method is simple and inexpensive, but the hard abrasives are easily detached from the surface of the iron substrate. The sintering method [20] is a method in which iron powder and a hard abrasive are mixed and then sintered, crushed, and sieved to obtain magnetic abrasives of the desired particle size. The hard abrasive of the magnetic abrasive produced by this method is encapsulated, and the iron matrix is prone to fragmentation during the magnetic abrasive finishing process, while the magnetic permeability of the magnetic abrasive is low.

The plasma spraying method [21,22,23] is a method of manufacturing spherical magnetic abrasives by high-temperature spherization of a plasma torch. The magnetic abrasives produced by this method have two insurmountable disadvantages: one is that the sharp cutting edge of the hard abrasive is passivated by the high temperature of the plasma torch and the other is that the hard abrasive is not firmly bonded to the iron substrate. The in situ alloy hardening method [24] is a magnetic abrasive preparation method that uses a chemical reaction to produce a hardened layer on the surface of the iron matrix. The cutting edge hardness of the magnetic abrasives produced by this method is low, making it difficult to machine materials of greater hardness. The chemical plating method [25] is a method of manufacturing magnetic abrasive preparation by co-depositing diamond powder and Ni-P alloy using a chemical plating solution. However, this method is a difficult way to produce magnetic abrasives with regular profiles, which leads to difficulty in maintaining the cutting edge of hard abrasives that leak out and which leads to low grinding efficiency. The atomization rapid coagulation method [26] is a magnetic abrasive preparation method based on the preparation of metal powder by atomization. Although the magnetic abrasives prepared by this method have the characteristics of long service life, high bond strength, high sphericity, etc., its atomization-based powder preparation method leads to less than half of the available particle size, which is more wasteful when using expensive hard abrasive powders such as diamond and cubic boron nitride.

At present, we have successfully prepared three kinds of magnetic abrasives, iron-based alumina [27], iron-based diamond [28], and iron-based cubic boron nitride [29], by using the plasma molten metal powder and powder jetting method with exciting results. The prepared magnetic abrasives were successfully applied in the inner wall polishing of ultra-fine and long vascular stent tubing, etc. [30,31]. At present, for the prepared iron-based alumina magnetic abrasives, the differences in microscopic morphology, physical phase composition, magnetic permeability, particle size distribution, and finishing ability between different particle sizes remain unexplored. For this reason, in this paper, two kinds of iron-based alumina magnetic abrasives with a range of particle size distributions were prepared, and their microscopic morphology, physical phase composition, magnetic permeability, particle size distribution, and finishing ability were characterized and compared, which provide a basis for the popularization and application of magnetic abrasive finishing in the industrial field.

## 2. Experiments

### 2.1. Synthetic Principle

The schematic diagram of the preparation of iron-based alumina magnetic abrasive is shown in Figure 1. The preparation equipment of the plasma molten metal powder and powder jetting method consists of a plasma torch generator, a hard abrasive spraying disk, precision powder feeders (one for a metal powder and one for a hard abrasive powder), a magnetic abrasive synthesizing and condensing chamber, a gas station, a dust collector, and a powder-collecting tank. The principle of the method is that the use of a plasma torch set to a particle size distribution in a narrow range of metal powder heated to a molten state, during the process of molten metal micro-droplets moving downward, leads to a hard abrasive high-speed shot to the metal micro-droplets, the formation of rapid condensation of the particle size distribution in a narrow range, a high bonding strength, a long service life, and a high degree of sphericity of the magnetic abrasive. The method not only overcomes the atomization method of magnetic abrasives prepared by less than half the range of the available particle size and the plasma spraying method of magnetic abrasives prepared by the sharp cutting edge blunt, overcoming all the combined shortcomings of those methods, but also combines the advantages of both methods. The prepared magnetic abrasives are characterized by a narrow particle size distribution range, strong bonding, a strong grinding ability, long service life, and high economy.

### 2.2. Composites Preparation

The raw materials used for the preparation of iron-based alumina magnetic abrasives by the plasma molten metal powder and powder jetting method were alumina powder (99.9% purity, d_50_ = 14 μm, Shandong Luxin High-tech Industry Co., Ltd., China) and spherical iron-based matrix powder (99.9% purity, a large particle size range of 106–120 μm and a small particle size range of 45–58 μm, Nangong Lijia Metal Material Co., Ltd., China). The parameters for the preparation of iron-based alumina magnetic abrasives are shown in Table 1. The steps to prepare the iron-based alumina magnetic abrasive are as follows: 1. put the raw materials into the powder feeder and turn on the dust collector; 2. turn on the working gas valve and the power supply and adjust the parameters of the equipment to ensure the stable operation of the plasma torch; 3. turn on the power supply of the powder feeder to start the preparation of the iron-based alumina magnetic abrasive; 4. turn off the working gas and the power supply and collect the magnetic abrasive; and 5. turn off the dust collector.

### 2.3. Composites Characterization

The microscopic morphology of the iron-based alumina magnetic abrasives prepared by the plasma molten metal powder and powder jetting method was characterized by scanning electron microscopy (SEM, Quanta 250 FEI, Hillsboro, OR, USA). The physical phase composition of the prepared iron-based alumina magnetic abrasives was analyzed by X-ray diffraction. The magnetic properties of the iron-based alumina magnetic abrasives were characterized using a vibrating sample magnetometer (VSM. Lake Shore e8600, Westerville, OH, USA). The range of particle size distribution of the iron-based alumina magnetic abrasives was investigated using a laser particle size analyzer (Microtrac, S3500, Montgomeryville, PA, USA). The MAF machining of AZ31B alloy was carried out on a CNC milling machine (XK7136C, Shandong Lunan Machine Tool Equipment Co., Ltd., Tengzhou, China), and the microscopic morphology and roughness of the workpiece surface before and after machining were evaluated by a DSX1000 digital microscope (OLYMPUS, Tokyo, Japan) and a probe roughness tester (TR200, Beijing Times High Tech Co., Ltd., Beijing, China), respectively.

## 3. Results and Discussion

### 3.1. Microstructure

Figure 2 shows the microscopic morphology of the iron-based alumina magnetic abrasive prepared by the plasma molten metal powder and powder jetting method. The diameter of the magnetic abrasive was measured to be about 160 μm in Figure 2a and about 70 μm in Figure 2c. From Figure 2a,c, we can see that the prepared iron-based alumina magnetic abrasive is of a regular spherical appearance and is not broken and that the spherical appearance can ensure the maximum cutting edge exposure and participation in processing, improving the polishing efficiency, as opposed to an irregular appearance of the magnetic abrasive where the hard abrasive is buried or cannot be exposed and it cannot employ the entire cutting edge in the processing, reducing the efficiency of polishing. An aluminum oxide and a hard abrasive were uniformly combined in the shallow surface layer of the iron substrate, of which part of the combination was with the iron substrate and the rest was exposed, so as to ensure that the combination of strength and cutting ability of both was buried in the substrate. In this way, the combined strength became high while the processing ability became greatly reduced, reducing the magnetic permeability of magnetic abrasives; in processing, magnetic abrasives are more prone to fly away from the processing area. From the further enlargement of Figure 2b,d, we can see that the alumina hard abrasive maintains a sharp cutting edge that has not been passivated. The alumina hard abrasives and iron substrates are tightly bonded and combined without gaps. By comparing the two grain sizes of the iron-based alumina magnetic abrasives, we found that the larger grain size of the iron-based alumina magnetic abrasive shallow surface layers combined with the alumina hard abrasives thicker than the grain size of the smaller abrasive. That is, the large grain size of the iron-based alumina magnetic abrasive is spherically better than the small grain size, which is due to the use of the same grain size of the hard abrasive.

### 3.2. Phase Composition

Figure 3 shows the XRD diagram of the iron-based alumina magnetic abrasives prepared by the plasma molten metal powder and powder jetting method. From the figure, it can be clearly seen that only the α-Fe phase and the Al_2_O_3_ phase are detected in the iron-based alumina magnetic abrasives with two particle size ranges, while the Fe_2_O_3_ phase is present. This indicates that during the preparation of magnetic abrasives by the metal powder plasma melt-spraying method, no oxidation occurs in the process of heat melting the iron matrix, resulting in bonding with the hard abrasive and in the subsequent cold solidification of the iron matrix, which ensures the bonding strength of the magnetic abrasives. The presence of the α-Fe phase also indicates that the iron matrix alumina magnetic abrasives have a high magnetic permeability, which ensures that they are not thrown out of the processing area by the centrifugal force during the finishing of the magnetic particles.

### 3.3. Magnetic Properties

Figure 4 shows the hysteresis loop diagram of the iron-based alumina magnetic abrasives prepared by the plasma molten metal powder and powder jetting method. From the figure, we can clearly see that, during the preparation of the two particle size ranges of the iron-based alumina magnetic abrasive, the hysteresis line was a thin narrow “S” shape, with the permeability of the two lower than the spherical iron substrate raw materials and with the large particle size range of the iron-based alumina magnetic abrasive permeability being slightly lower than that of the small particle size range. The reason for this situation is that the alumina hard abrasive combined in the surface layer of the iron matrix is considered to affect the magazines of the magnetic domains of the iron matrix, resulting in a reduction in the magnetic phase of the magnetic abrasive; in other words, being combined in the superficial layer of the iron matrix of the hard abrasive impurities results in a lower magnetic permeability.

### 3.4. Particle Size Distribution

Figure 5 shows the particle size distribution of the iron-based alumina magnetic abrasives prepared by the plasma molten metal powder and powder jetting method. From the figure, we can see two particle size ranges of the magnetic abrasives beyond the range of the particle size distribution of the iron powder raw materials distributed, respectively, between 100~130 μm and 40~70 μm. There are two main reasons for this change in particle size: The first reason is that the alumina hard abrasive and the molten iron matrix combined and cooled to form a magnetic abrasive, due to the presence and impact of the alumina hard abrasive on the magnetic abrasive particle size followed by an increase in the iron substrate. The second reason is that the high-speed alumina hard abrasive will interact with the molten iron substrate, caused by a certain degree of impact, so that the iron substrate particle size becomes smaller. The two reasons together explain why the magnetic abrasive particle size distribution has changed somewhat. From the figure, we also found that the two types of magnetic abrasives remained in the size of 109.26 μm and 50.12 μm, respectively; no larger particle size or smaller particle size of the magnetic abrasives exist. The reason for the absence of larger-sized magnetic abrasives is that the raw materials used in the iron matrix do not have the ability to attract each other, while the appropriate preparation process parameters also play a role. The absence of smaller-sized magnetic abrasives is mainly due to the fact that the smaller-sized magnetic abrasives are removed by the dust collector.

### 3.5. Finishing Properties

The grinding performance test of the prepared magnetic abrasives was carried out by using XK7136C with the spindle retrofitted with planar slotted magnetic poles (2 mm × 2 mm × 60°) as shown in Figure 6; the test parameters are shown in Table 2.

According to the process parameters shown in Table 2, the AZ31B alloy was grounded, and the surface roughness values were measured every 3 min, and the surface roughness of the workpiece over time was obtained as shown in Figure 7. It can be seen from the figure that the workpiece surface roughness with time shares the same changing trend in both of the two process parameters: in the first 6 min, the workpiece surface roughness decreased sharply; from 6 min to 12 min, the workpiece surface roughness changes are more gentle and reached the lowest value; after 12 min, the workpiece surface roughness increased slowly. This is due to the fact that in the initial stage of machining, the bumps on the surface of the workpiece are first removed, and, as the machining time increases, the surface roughness of the flat surface is scratched out of the pits by the cutting edge of the magnetic abrasive, causing the surface roughness of the workpiece to increase. We can also see from the figure that the surface roughness of the workpiece can be reduced after grinding with smaller magnetic abrasives, and, after 12 min of machining, the surface roughness of the workpiece decreased from 485 nm to 98 nm.

A comparison of the surface topography of the workpiece obtained before and after 12 min of processing with a small-sized magnetic abrasive is shown in Figure 8. From the figure, we can see that after 12 min of processing, the deeper scratches on the surface of the workpiece are removed and the raised peaks are basically removed; the two-dimensional contour of the workpiece surface tends to flatten from the violent jump before processing.

## 4. Conclusions

In this paper, two kinds of iron-based alumina magnetic abrasives with different particle sizes were prepared by the plasma molten metal powder and powder jetting method, and their microscopic morphology, physical phase composition, magnetic permeability, particle size distribution, and abrasive ability were characterized, and the following conclusions were obtained from the study:(1)The hard abrasive bound to the surface layer of the magnetic abrasive with smaller particle sizes is sparser than that of the magnetic abrasive with larger particle sizes;(2)Both magnetic abrasives are composed of α-Fe and Al_2_O_3_;(3)The magnetic permeability of the smaller magnetic abrasive is slightly higher than that of the larger one;(4)Both magnetic abrasives are distributed in the smaller particle size range;(5)Lower surface roughness can be obtained by using the smaller particle size magnetic abrasive for MAF.

## Figures and Tables

**Figure 1 micromachines-15-00709-f001:**
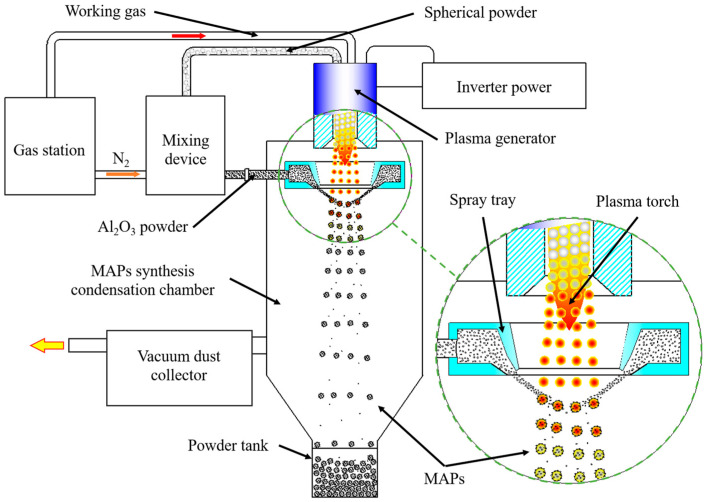
Principle diagram of the preparation of iron-based Al_2_O_3_ magnetic abrasives.

**Figure 2 micromachines-15-00709-f002:**
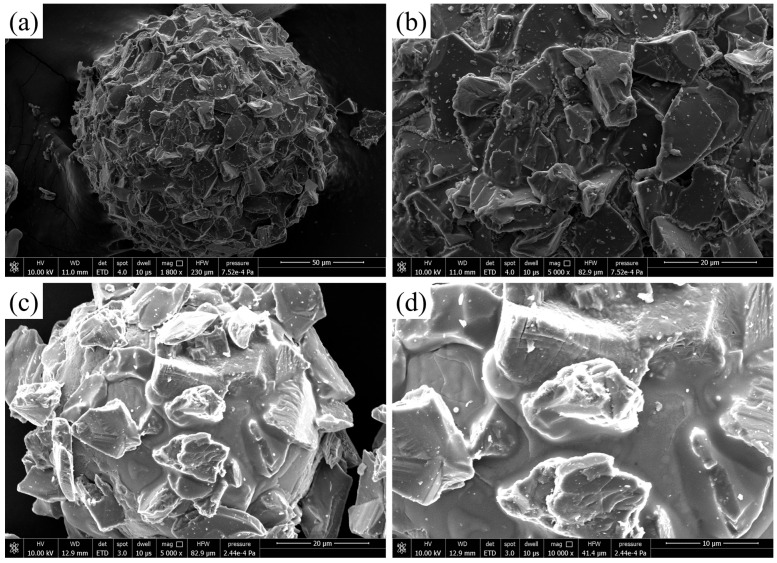
Microscopic morphologies of iron-based Al_2_O_3_ magnetic abrasives: (**a**) overall view of large-grained magnetic abrasives; (**b**) localized view of large-grained magnetic abrasives; (**c**) overall view of small-grained magnetic abrasives; (**d**) localized view of small-grained magnetic abrasives.

**Figure 3 micromachines-15-00709-f003:**
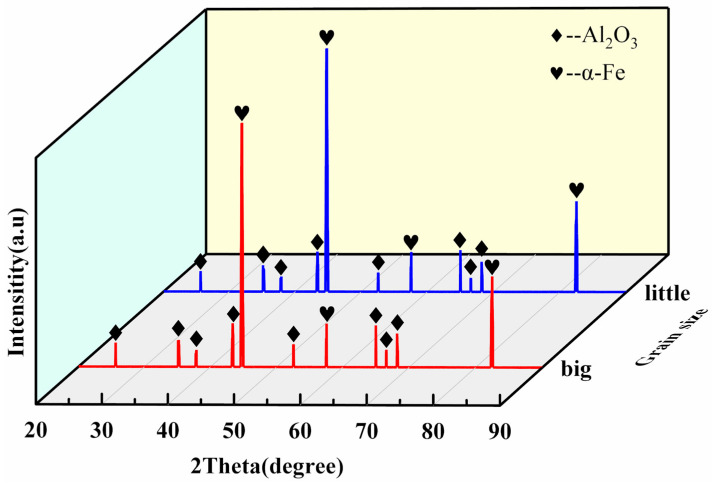
XRD pattern of iron-based alumina magnetic abrasives.

**Figure 4 micromachines-15-00709-f004:**
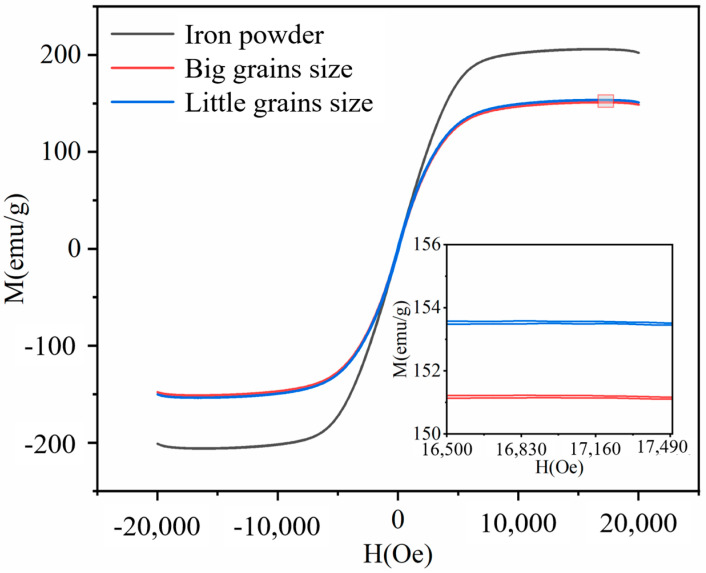
Hysteresis loop diagram of iron-based alumina magnetic abrasives.

**Figure 5 micromachines-15-00709-f005:**
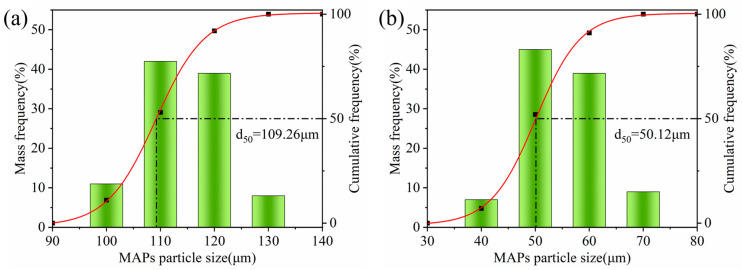
Particle size distribution of iron-based alumina magnetic abrasives. (**a**) Big MAPs particle size. (**b**) Little MAPs particle size.

**Figure 6 micromachines-15-00709-f006:**
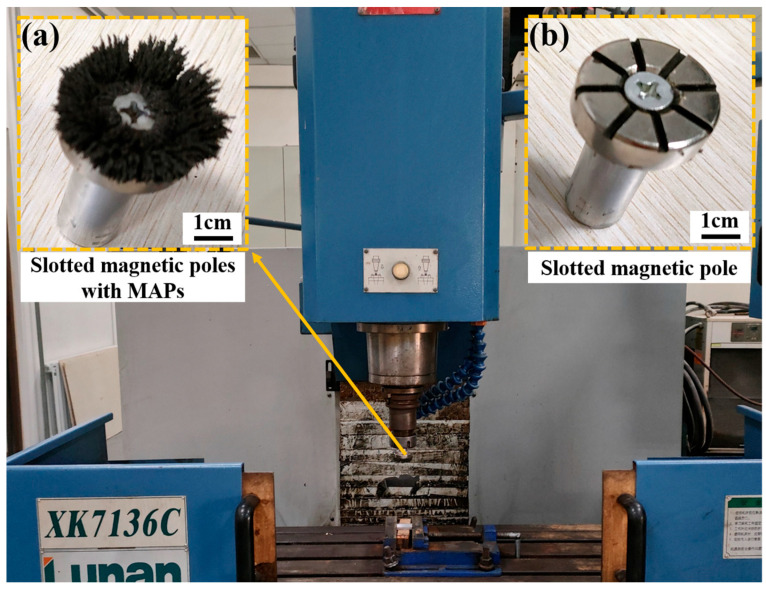
Diagram of the experimental device: (**a**) slotted magnetic poles with MAPs; (**b**) slotted magnetic poles.

**Figure 7 micromachines-15-00709-f007:**
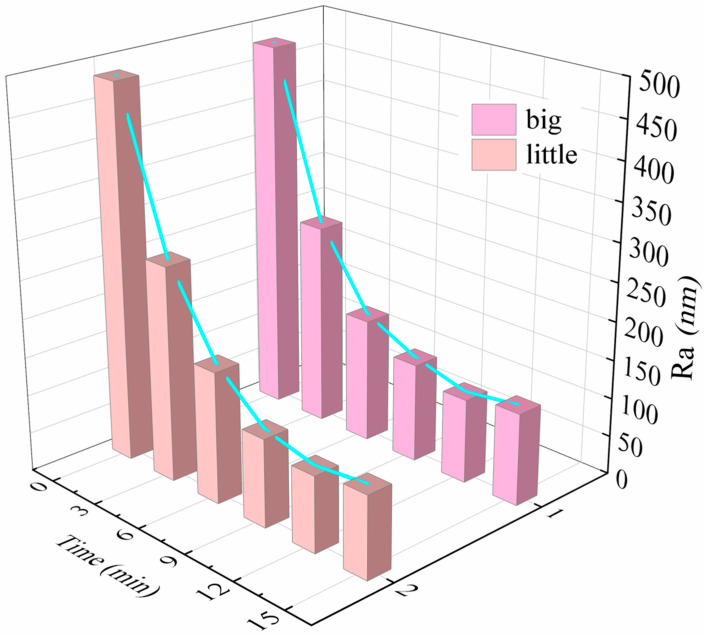
Plot of workpiece surface roughness over time.

**Figure 8 micromachines-15-00709-f008:**
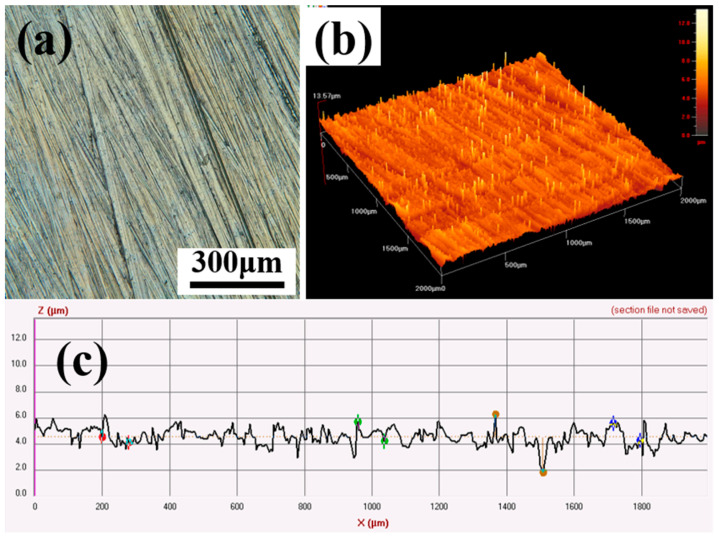
Comparison of the surface morphology before and after machining AZ31B with small magnetic abrasives: (**a**) microphotomorphology before machining; (**b**) three-dimensional phantom morphology before machining; (**c**) surface contour lines before machining; (**d**) microphotomorphology after machining; (**e**) three-dimensional phantom morphology after machining; (**f**) surface contour line after machining.

**Table 1 micromachines-15-00709-t001:** Operating conditions experiments.

Nozzle	Ring Seam
Nozzle cone angle (°)	65
Nozzle annular seam diameter (mm)	3.5
Nozzle bore diameter (mm)	46
Inlet pressure of nozzle (MPa)	0.8
Distance between nozzle and plasma generator (mm)	70
I (A)	700
Ar (L/h)	1000
H_2_ (L/h)	210
Iron powder (g/min)	40
Iron powder particle size (μm)	106~120, 45~58
Al_2_O_3_ powder (g/min)	240
Equipment power (kW)	25.34

**Table 2 micromachines-15-00709-t002:** Finishing parameters of the experiments.

No.	Magnetic Abrasive Grain Size	Spindle Speed (rpm)	Gap (mm)	Magnetic Abrasive Use Quality (g)
1	Big	1500	2	2
2	Little			

## Data Availability

Data is contained within the article.

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
