# Peer review of "Investigation of Synthesis, Characterization, and Finishing Applications of Spherical Al2O3 Magnetic Abrasives via Plasma Molten Metal Powder and Powder Jetting"

_micromachines, 2024, doi:10.3390/mi15060709_

Round 1

Reviewer 1 Report

Comments and Suggestions for Authors

This paper used the plasma molten metal powder and powder jetting method to prepare magnetic abrasives of various particle sizes. And then their physical properties and processing performance were characterized, which has certain engineering application value. However, the article still has the following issues that need to be addressed:

(1) In the Introduction, the paper used various terms, such as magnetic abrasive finishing, magnetic particle grinding, and magnetic particle milling. However, finishing, grinding, and even milling refer to different machining processes. Please use a consistent term based on the actual machining process.

(2) In Section 2.2, the author discussed the preparation process. In the fourth step, the collection of magnetic abrasives was mentioned. However, when using the plasma molten metal powder and powder jetting method, is there a possibility that the magnetic abrasives might be mixed with pure iron powder or pure abrasive? How are qualified products and waste materials accurately screened?

(3) The font size in the figures throughout the paper should be consistent. For example, the font size in Figure 3 is clearly inconsistent with other figures.

(4) The descriptions of Figures 7 and 8 lack explanations of the phenomena. It is suggested to add comprehensive explanations.

(5) The ordinate name is missing in Figure 7.

(6) The Conclusion section should focus on the main findings of the research work, avoiding a recapitulation of the work discussed earlier. Please improve this section.

(7) The format of the references should be consistent. Some list all authors' names, while others use "et al." for abbreviation.

(8) Please enhance the English language expression throughout the paper.

Comments on the Quality of English Language

Please enhance the English language expression throughout the paper.

Author Response

  1. In the Introduction, the paper used various terms, such as magnetic abrasive finishing, magnetic particle grinding, and magnetic particle milling. However, finishing, grinding, and even milling refer to different machining processes. Please use a consistent term based on the actual machining process.

Answer: We are very sorry for our negligence of this section. I have revised my manuscript, and the specific revised parts have been marked in red in the submitted manuscript.

  1. In Section 2.2, the author discussed the preparation process. In the fourth step, the collection of magnetic abrasives was mentioned. However, when using the plasma molten metal powder and powder jetting method, is there a possibility that the magnetic abrasives might be mixed with pure iron powder or pure abrasive? How are qualified products and waste materials accurately screened?

Answer: Thank you for your comments on this section. When using plasma molten metal powder and powder jetting methods, magnetic abrasives are mixed with smaller portion of unbound pure iron powder and hard abrasives. Hard abrasives can be obtained by sieving and reused. Pure iron powder and magnetic abrasives are more difficult to separate because of the overlap between their particle size distributions.

  1. The font size in the figures throughout the paper should be consistent. For example, the font size in Figure 3 is clearly inconsistent with other figures.

Answer: We are very sorry for our negligence of this section. I have revised my manuscript, and the specific revised parts have been marked in red in the submitted manuscript.

  1. The descriptions of Figures 7 and 8 lack explanations of the phenomena. It is suggested to add comprehensive explanations.

Answer: We are very sorry for our negligence of this section. I have revised my manuscript, and the specific revised parts have been marked in red in the submitted manuscript.

  1. The ordinate name is missing in Figure 7.

Answer: We are very sorry for our negligence of this section. I have revised my manuscript, and the specific revised parts have been marked in red in the submitted manuscript.

  1. The Conclusion section should focus on the main findings of the research work, avoiding a recapitulation of the work discussed earlier. Please improve this section.

Answer: Thank you for your comments on this section. I have revised my manuscript, and the specific revised parts have been marked in red in the submitted manuscript.

  1. The format of the references should be consistent. Some list all authors' names, while others use "et al." for abbreviation.

Answer: We are very sorry for our negligence of this section. I have revised my manuscript, and the specific revised parts have been marked in red in the submitted manuscript.

  1. Please enhance the English language expression throughout the paper.

Answer: Thank you for your comments on this section. We have polished the paper. I have revised my manuscript, and the specific revised parts have been marked in red in the submitted manuscript.

Reviewer 2 Report

Comments and Suggestions for Authors

Authors: Shujun Wang1*, Yusheng Zhang2, Shuo Meng2, Yugang Zhao2 and Jianbing Meng2

Article title: Investigation of synthesis, characterization and finishing applications of spherical Al2O3 magnetic abrasives plasma via molten metal powder and powder jetting

Currently, high demands are placed on the processing of complex surfaces of difficult-to-machine parts with high hardness, which are used in the medical, aerospace, military, transportation and other industries. To some extent, the problem of processing such complex parts can be solved by reducing their roughness, possibly using magnetic abrasive processing technology. Magnetic abrasives are used as tools for magnetic abrasive finishing and their performance characteristics affect the efficiency and quality of processing. This study examines the production and properties of iron-based alumina magnetic abrasives with a specific particle size distribution range. The grinding and polishing abilities, magnetic permeability, morphology and phase composition of magnetic abrasives are characterized and compared.

Notes:

1. In section 3.1 Microstructure, it is necessary to indicate the dimensions of the presented magnetic abrasives based on aluminum oxide and give their quantitative characteristics.

2. The authors point to the adhesion strength of abrasives. Is this conclusion made only based on the results of operation, or are there physical parameters that determine the indicated strength?

3. Was the amount of penetration of the abrasive into the base determined, and what parameters of the method affect and can be adjusted to obtain the required best result for longer-term operation of the magnetic abrasive?

4. Indicate the temperature of plasma melting of metal powder at which, as the authors indicate, oxidation of the iron matrix does not occur during thermal melting and the Fe2O3 phase is absent in the X-ray diffraction pattern.

5. In conclusion, it is necessary to give practical recommendations for the most suitable magnetic abrasives from the point of view of particle size distribution, magnetic permeability and operating properties during grinding and polishing.

Paper can be accepted after specified adjustments

Author Response

  1. In section 3.1 Microstructure, it is necessary to indicate the dimensions of the presented magnetic abrasives based on aluminum oxide and give their quantitative characteristics.

Answer: We are very sorry for our negligence of this section. I have revised the manuscript to add a quantitative description of the magnetic abrasive diameter, and the specific revised parts have been marked in red in the submitted manuscript.

  1. The authors point to the adhesion strength of abrasives. Is this conclusion made only based on the results of operation, or are there physical parameters that determine the indicated strength?

Answer: Thank you for your comments on this section. As far as the bond strength of magnetic abrasives is concerned, according to the currently known references, it is not possible to measure the bond strength of an iron substrate to a hard abrasive by direct methods, but it can usually only be analyzed by means of magnetic grain grinding experiments. The bond strength between the iron matrix and the hard abrasive can be qualitatively analyzed from the microscopic morphology.

  1. Was the amount of penetration of the abrasive into the base determined, and what parameters of the method affect and can be adjusted to obtain the required best result for longer-term operation of the magnetic abrasive?

Answer: Thank you for your comments on this section. This is a very good research direction for the magnetic abrasive preparation method described herein, and we will discuss which parameters affect the bond depth of the hard abrasive with reference to the grinding capacity and service life of the magnetic abrasive through the next research.

  1. Indicate the temperature of plasma melting of metal powder at which, as the authors indicate, oxidation of the iron matrix does not occur during thermal melting and the Fe2O3 phase is absent in the X-ray diffraction pattern.

Answer:  Thank you for your comments on this section. In the magnetic abrasive preparation method described in this article, the iron powder is melted under the protection of inert gas, and Fe2O3 is not produced.

    5. In conclusion, it is necessary to give practical recommendations for the most suitable magnetic abrasives from the point of view of particle size distribution, magnetic permeability and operating properties during grinding and polishing.

Answer: Thank you for your comments on this section. I have revised my manuscript, and the specific revised parts have been marked in red in the submitted manuscript.